# Establishment of Personalized Finite Element Model of Crystalline Lens Based on Sweep-Source Optical Coherence Tomography

**Guangheng Liu** [1], **Ang Li** [2], **Jian Liu** [1,3,*], **Yuqian Zhao** [1], **Keliang Zhu** [1], **Zhen Li** [1], **Yang Lin** [1], **Shixin Yan** [1], **Hongyu Lv** [4], **Shuanglian Wang** [5], **Yao Yu** [1,3], **Yi Wang** [1,3], **Jingmin Luan** [6] and **Zhenhe Ma** [1,3]

[1] School of Control Engineering, Northeastern University at Qinhuangdao, Qinhuangdao 066004, China
[2] Hubei Aerospace Flight Vehicle Institute, Wuhan 430040, China
[3] Hebei Key Laboratory of Micro-Nano Precision Optical Sensing and Measurement Technology, Qinhuangdao 066004, China
[4] Department of Ophthalmology, Qinhuangdao Maternal and Child Health Hospital, Qinhuangdao 066004, China
[5] Department of Ophthalmology, Tangshan Maternal and Children Hospital, Tangshan 063000, China
[6] School of Electronic Computer and Communication Engineering, Northeastern University at Qinhuangdao, Qinhuangdao 066004, China
\* Correspondence: liujianneuq@163.com

**Abstract:** The virtual lens model has important value in ophthalmic research, clinical diagnosis, and treatment. However, the establishment of personalized lens models and the verification of accommodation accuracy have not been paid much attention. We proposed a personalized lens model establishment and the accommodation accuracy evaluation method based on sweep-source optical coherence tomography (SS-OCT). Firstly, SS-OCT is used to obtain a single lens image in the maximum accommodation state. After refraction correction, boundary detection, and curve fitting, the central curvature radius, thickness, and lens nucleus contour of the anterior and posterior surfaces of the lens were obtained. Secondly, a personalized finite element model improved from Burd's model was established using these individual parameters, and the adaptation process of the lens model was simulated by pulling the suspensory ligament. Finally, the shape and refractive power changes of the real human lens under different accommodation stimuli were collected and compared with the accommodation process of the finite element model. The results show that the accommodation process of the finite element model is highly consistent with that of the real lens. From the un-accommodation state to the maximum-accommodation state, the difference rate of all geometric and refractive parameters between the two is less than 5%. Thus, the personalized lens finite element model obtained by the calibration and correction of the existing model can accurately simulate the regulation process of a specific human lens. This work helps to provide a valuable theoretical basis and research ideas for the study of clinical diagnosis and treatment of related diseases.

**Keywords:** sweep-source optical coherence tomography; finite element modeling; lens accommodation; accommodative response

## 1. Introduction

The research of virtual organs is a very popular field in biomedical engineering. The virtual eye model has important value in ophthalmic research, clinical diagnosis, and treatment [1]. With the development of science and technology, computer and information technology play an increasingly important role in the research of medicine and biology. Using information technology to realize the digitalization and visualization of the structure and function of the human body from the micro to the macro, and finally to achieve the accurate simulation of the human body will play an important role in the

development of medicine and biology. The lens is an important part of the human eye and the core of the regulatory system. Myopia, presbyopia, cataract, and other common diseases are directly related to the lens [2]. Therefore, modeling a human lens can provide important guidance for the research, clinical diagnosis, and treatment of related diseases. For example, the lens model can be used to study the regulatory ability of the lens and help reveal the pathogenesis of presbyopia, to find an effective treatment [3,4]. The lens model can also be used to simulate the main visual functions of the human eye, such as refraction and imaging, and to study the changes in the refractive system and the effects of diseases on the eyes [5–7]. Another purpose of the human eye lens model is to calculate the lens diopter before cataract surgery or perform a surgical simulation to reduce the risk [8].

Lens modeling has been performed in many different ways, such as Burd using polynomial plus arc fitting [9], Chung using piecewise polynomial fitting [10], Kasprzak using hyperbolic cosine function fitting [11], and Zhang et al. implementing elliptic curve fitting with MATLAB [12]. In the regulation of lens deformation, Weale and Wyatt established a simple mechanical model to simulate the deformation of the lens [13,14]. The electronic medical engineering laboratory of Xiamen University established a parametric model of the lens based on multi-source lens data fusion [15]. Based on the established morphology of the lens and capsule of the Chinese human eye, a spring model was designed to realize the mechanical simulation of the capsulorhexis of the lens capsule [16].

The rapid development of the finite element method, an efficient universal numerical technique, began in the 1950s with the growing use of electronic computers. It has been widely used in solid mechanics, fluid mechanics, genetics, electromagnetism, biomechanics, and other mathematical and physical fields [17]. With the great progress of computer performance and numerical simulation technology, the finite element method has been widely used in the optical surfaces modeling of the eye, such as Sanchez et al. creating ten patient-specific finite element models to estimate the strain and stress fields in the cornea in preoperative and postoperative configurations [18]. Amini et al. created a finite element model of the anterior segment that included changes in the iris contour and the aqueous humor flow, and the contact distance between iris and lens and the change of intraocular pressure under various iris root rotations are simulated [19]. Tse et al. used finite element models and simulations of this surgical procedure were used to investigate the relative effects of various shapes and dimensions of scleral flap and sclerostomy on the aqueous outflow [20]. Wang et al. proposed a three-dimensional finite element model to simulate the viscoelastic interaction between the microrobot and porcine vitreous humor, which successfully validated the experimental measurements [21]. Norman created a finite element modeling to predict the effects of pressurizing the eyes to an iOP of 30 mmHg, with the results used to characterize the effect of inter-individual differences in scleral dimensions on the biomechanics of the optic nerve head [22]. For the modeling and simulation of the lens accommodation process, the most classic is the finite element model established by Burd using ABAQUS software [9]. On this basis, Heiner established a finite element model that conforms to the Coleman and Helmholtz adjustment theory [23]. Lanchares developed a finite element model of a 30-year-old lens in the accommodated state to determine age-dependent change in the material properties of the tissues composing the human crystalline lens [24]. Epigg presented a biomechanical eye finite model to induce pseudophakic accommodative movement for the evaluation of the focal shift of accommodative intraocular lenses, which indicated that the vitreous plays an important role for the functionality of accommodative iOLs [25].

The correctness of the majority of the data utilized in the current generic lens models, which comes from previously published datasets or the average data of many subjects, is hotly debated. Although corneal topography and axial length are similar, different human eyes may have different lens morphology and axial position [7,26]. Factors, such as race, gender, and myopia, can also affect the shape of the lens. Even in the same individual, the lens biological characteristic parameters of different ages are also very different.

Improving the accuracy of model adjustment is crucial for disease diagnosis, diopter research, optimal design of the intraocular lens, and surgical treatment of diseases. Therefore, personalized modeling of the lens adjustment process for specific groups is crucial. Personalized lens modeling can not only improve the accuracy of disease diagnosis and surgical simulation   but also help guide the design of intraocular lens (iOL) structure, so that it can not only correct defocus and astigmatism but also correct spherical aberrations, thereby improving the visual function of the human eye [27–29]. At present, personalized modeling technology has achieved remarkable research results in the pulmonary artery, liver pipeline, and artificial bone [30–32]. However, the outcomes of research on personalized lens models with precise accommodation capabilities have not typically been published.

There are two main challenges faced in personalized lens modeling with accurate accommodation ability. The first is to obtain the complete parameters of the individual lens as accurately as possible. The second is to select a suitable deformation parameter to ensure that the model can accurately simulate the accommodation process of the real lens. Commonly used lens parameters acquisition equipment mainly include Scheimpflug camera, slit lamp, ultrasound imaging, magnetic resonance imaging (MRI), etc. [33–38]. The resolution of Scheimpflug cameras is low and susceptible to impurities [39]. A slit lamp can obtain clear lens images, but the refraction of the cornea and aqueous humor will bring errors to the measurement [33]. Ultrasound and MRI can avoid the above shortcomings, but their resolution is too low to accurately measure the lens deformation during accommodation [36,38].

Due to the limitations of these methods, Optical coherence tomography (OCT) has been proposed. OCT is a technology that was developed in the early 1990s for ophthalmological applications and is commonly used [40,41]. OCT is the optical equivalent of ultrasound, using light instead of sound to produce images of tissue. Resolutions up to 1–2 μm can be achieved, being 100–250 times higher than high-resolution ultrasound [42] and approaching that of microscopy. However, due to light scattering by the sample, imaging depth is usually limited to 1~2 mm. An image produced by OCT resembles the tissue architecture observed in histology and can, therefore, be considered as an "optical biopsy". Since 1994, the first eye image was taken by Izatt et al., OCT gradually became the most commonly used equipment for ophthalmology diagnosis of the cornea, marginal angle, and lens [43].

Within a decade, OCT completed the transformation from the time domain OCT (TD-OCT) to the frequency domain OCT (FD-OCT) [44]. Frequency domain OCT includes spectral-domain OCT (SD-OCT) and sweep-source OCT (SS-OCT) [45]. The swept-source OCT (SS-OCT) developed from basic OCT technique has improved imaging speed, greater depth, and range of structure visualization [46–49], which allows for improvements in OCT-imaging in the clinic of ophthalmology [50–52]. SS-OCT also has been successfully used to measure ciliary muscle size in a large number of healthy subjects. Therefore, SS-OCT represents a promising tool for the monitor of the lens accommodation process [53].

In this paper, we proposed a personalized lens finite element modeling method based on Burd's model. SS-OCT is used to obtain a single lens image under the maximum adjustment state. After algorithm processing, such as refraction correction, boundary detection and curve fitting, and the center curvature radius of the anterior and posterior surface, the thickness and the nucleus contour of the lens were obtained. A personalized finite element model was established using these individual parameters, and the accommodation process of the lens was simulated by pulling the suspensory ligament. We also collected the shape and refraction power changes of real human lens under different accommodation stimulus by SS-OCT. The accommodation accuracy of the finite element model was verified by comparing them.

## 2. Methods

### 2.1. Subjects

In this study, we selected 3 subjects with healthy lenses and imaged the left eye lens of each subject. Their ages ranged from 23 to 29 years, and diopters ranged from 0 D to 2.00 D. Subjects did not require refractive correction. Subjects have no history of eye disease, trauma, or any eye surgery. The research followed the tenets of the Declaration of Helsinki and was approved by the Ethics committee of Northeastern University (No. Neu-ec2020a009s). Participants were informed of the nature of the experiments and provided written consent.

### 2.2. Measurement System and Data Acquisition

In this paper, the human-built SS-OCT system with a central wavelength of 1310 mm and a bandwidth of 100 nm is used to image the lenses of three subjects. Figure 1 is the schematic diagram of the SS-OCT system. The light source employed is an akinetic swept-source (MEMS-VCSEL, Thorlabs Inc., Newton, NJ, USA), operating at 200 kHz swept rate and an axial resolution of approximately 7.5 μm in air. The infrared light emitted by the sweep frequency light source is transmitted to the reference and detector arm through a single-mode optical fiber coupler. The beam emitted by the swept source was split into the sample arm and the reference arm by a 50:50 ratio coupler (PDB480C-AC, Thorlabs). In the sample arm, an aiming beam was combined with another 50:50 coupler to guide the OCT imaging. The sample light passes through the lens L1 with a focal length of 150 mm and converges with the beacon light to form a beam entering the human eye, to achieve the lateral resolution of approximately 16 μm.

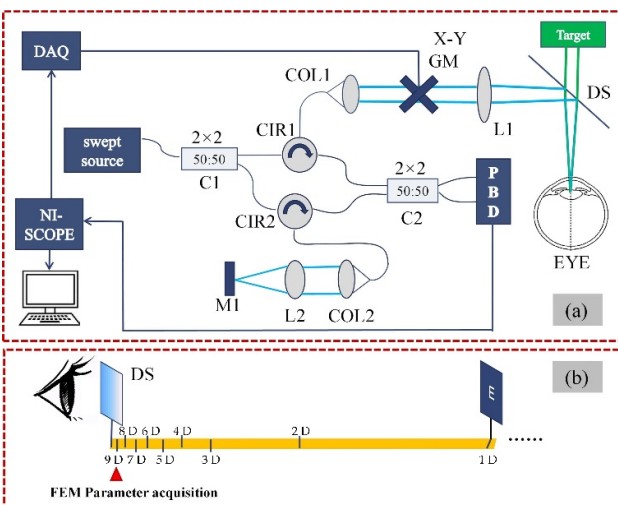

**Figure 1.** Schematic diagram of the SS-OCT system (**a**) and the acquisition of finite element model parameter (**b**). L1, L2: lens; C1, C2: fiber coupler; PBD: balanced photodetector; M1: mirror; COL1, COL2: collimator; CIR1, CIR2: circulator; GM: galvanometric mirror. DS: dichroic mirror.

Each B-Scan contains 600 A-Scans, and each A-Scan contains 3500 pixels to cover the 25 mm imaging depth. The entire front segment can be fully rendered in one scan. The laser power on the cornea is about 3 mW, which is far below the safety standard set by the American National Standards Institute. The back scattered light of the human eye and the reflected light of the reference mirror pass through the circulators CIR 1 and CIR 2 and interfere with the coupler C2. The OCT interferogram is detected using a 1.6 GHz bandwidth high-speed double balanced photodetector (Thorlabs, PDB480C-AC) and a 5 GHz sampling rate super data sampling digital card (PXLe-5162, National Instruments, Austin, TX, USA).

We placed a sliding guide rail at the "Target" in Figure 1a. A luminous beacon plate with the black letter "E" was placed on the guide rail. The letter height is 1.5 mm, as shown in Figure 1b. The device was used to observe the lens morphology in 10 accommodation states (0~9 D). Among them, the lens morphology under the maximum accommodation state (9 D) was used to construct the finite element model. By changing the traction force on the model, we simulate the morphological change of the lens arising from the accommodation stimulus. Other lens data under the accommodation state (0~8 D) were used to verify the accommodation accuracy of the personalized lens model. In the experiment, 50 human eye images were obtained each time the adjustment stimulus was changed. Among them, the 20 images with the highest similarity are selected and averaged, and the average image is used to calculate the eye parameters.

### 2.3. Image Processing

Figure 2a is an original image of the anterior segment obtained by SS-OCT. In order to extract parameter information, such as surface boundaries, central thickness, and curvature radius of the lens, a series of image processing techniques were used.

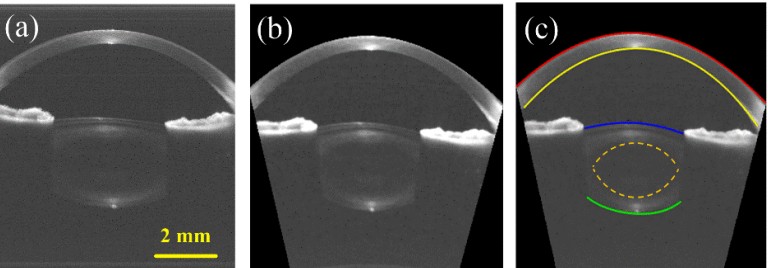

**Figure 2.** Boundary extraction of lens and nucleus. (**a**) Original image of the anterior segment obtained by SS-OCT. (**b**) Optical refractive correction. (**c**) Contour of the boundaries of the cornea and lens.

Refraction correction: Light will be refracted as it enters human eyes from the air. Therefore, it is necessary to correct the optical refraction of the original image after gray scale transformation and spatial filtering [54]. Pixel points located at the corneal anterior surface were extracted for curve fitting, and the curve equation was expressed by $f(j)$. The j is the abscissa of the image. The new coordinates of the corrected pixel point can be expressed as:

$$y_B(i,j) = f(j) + \cos[A_{in}(j) - A_{out}(j)] \times [i - f(j)] / n_{21} \tag{1}$$

$$x_B(i,j) = j - \tan[A_{in}(j) - A_{out}(j)] \times [y_B(i,j) - f(j)] \tag{2}$$

where $A_{in}(j)$ and $A_{out}(j)$ represent the incident angle and refraction angle, respectively. The ratio of the refractive index of medium 2 to medium 1 is represented by $n_{21}$. The corrected image was smoothed and gradient template filtering was used to highlight lens boundaries. Then, active contour model algorithm (Snake model) [55] was used to outline the anterior and posterior surface boundaries of the lens. Refraction correction results are shown in Figure 2b.

Boundary extraction and curvature calculation: We used vertical ascending gradient template and descending gradient template to enhance the boundary of cornea and lens front surface (gray value from low to high) and back surface (gray value from high to low). In this paper, we chose Canny operator as the boundary detection function after comparing the experimental methods [56]. Non maximum suppression method and threshold adjustment are adopted for the searched boundary to form a boundary curve that is relatively easy to identify and extract. After that, the contour equation of the lens surfaces was obtained by the fifth-degree polynomial fitting method in MATLAB based on the least square method [57]. The polynomial fitting expression is:

$$x(t) = c_0 + c_1 t + c_2 t^2 + c_3 t^3 + c_4 t^4 + c_5 t^5 \qquad (3)$$

$$v(t) = c_1 + 2c_2 t + 3c_3 t^2 + 4c_4 t^3 + 5c_5 t^4 \qquad (4)$$

$$a(t) = 2c_2 + 6c_3 t + 12c_4 t^2 + 20c_5 t^3 \qquad (5)$$

where, between two points, the start time of a trajectory planning is $t_0$ and the end time is $t_1$, the starting point is $(x_0, v_0, a_0)$, and the ending point is $(x_1, v_1, a_1)$. Generally, the starting point time is $t_0 = 0$ by default. According to the quintic polynomial and its first and second derivatives, the coefficients can be obtained: $c_0$, $c_1$, $c_2$. Take the obtained $c_0$, $c_1$, $c_2$ back to the original formula and its first and second derivatives to find $c_3$, $c_4$, $c_5$. The accurate results of the radius of curvature of the front and rear surfaces of the lens were shown in Figure 2c.

Due to the uneven distribution of proteins in the lens (different refractive indices), the human lens is generally divided into two main regions: the nucleus and the cortex. According to the experience obtained from previous studies [58], we selected the point with the maximum gradient value in the lens to generate a boundary between cortex and nucleus by the snake model. The contour equation of the lens nucleus was obtained by polynomial fitting. The maximum distance between the lens nucleus contours was defined as the nucleus thickness.

## 2.4. Finite Element Modeling

The lens finite element model consists of three parts: geometrical parameters, material parameters, and boundary conditions. The personalized lens finite element model described in this paper is improved based on the model proposed by Burd, Judge, and Cross et al. [9,59], as shown in Figure 3. In terms of geometric parameters, the model is composed of three parts: lens, suspensory ligament, and ciliary body. The lens consists of a capsule, cortex, and nucleus. To establish an accurate personalized lens model, we adjusted several key geometric parameters in the basic model.

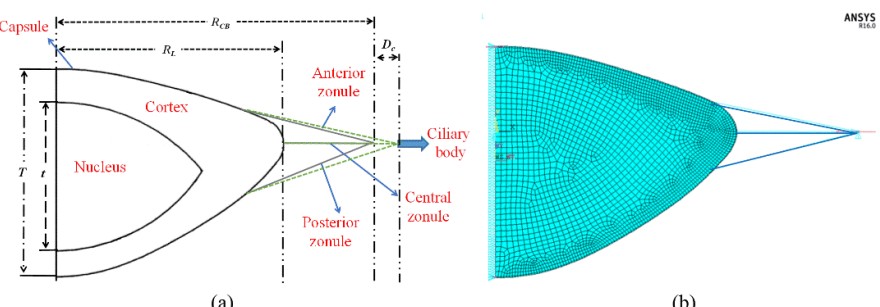

**Figure 3.** (**a**) Personalized lens geometric parameters. $Rcb$: Ciliary body radius; $RL$: Lens radius; $Dc$: Amplitude of radial movement of ciliary body; $T$: Lens thickness; $t$: Nucleus thickness; (**b**) Element diagram of finite element model in maximum accommodation state.

First, the contours of the anterior and posterior surfaces (visible parts) of the individual lens measured by SS-OCT were used as the geometric parameters of the lens profile in the finite element model. The measured individual lens nucleus was used as geometric parameters of the boundary between the cortex and nucleus.

Second, the complete geometry of the lens and ciliary body cannot be measured directly due to the occlusion of the iris. Fortunately, the regression polynomial of the lens and ciliary body diameters with age under the fully accommodated state has been proposed by previous studies [60]. Based on this, we estimated the approximate diameters of the lens and ciliary body according to the age of the subjects and use them as the diameter parameters in the personalized finite element model.

Third, in addition to the personalized geometry parameters, the material parameters of the finite element model of the lens also differs greatly with age. In this paper, we refer to the relevant laws of material parameters changing with age summarized by Krag et al. [9,61] and chose the material parameters that meet the age of the subject.

In this paper, the fifth-order polynomial was used to describe the lens profile under the maximum accommodation. To make the slope of the polynomial at the center line of the lens be 0, the coefficient of the first-order term of the fifth-order polynomial was set to 0 [39]. The polynomial can be expressed as $f(j) = aj^5 + bj^4 + cj^3 + dj^2 + e$. The remaining coefficients (a to d) were determined by least square fitting. The lens surface contour fitting parameters of the three subjects are shown in Table 1. Other parameters involved in finite element lens modeling are listed in Table 2.

**Table 1.** Polynomial parameters of lens surface profile.

| Parameter | $a$ | $b$ | $c$ | $d$ | $e$ |
|---|---|---|---|---|---|
| ALC (Subject 1) | −0.001 | 0.0106 | −0.0299 | −0.0686 | 2.2 |
| PLC (Subject 1) | −0.0001 | 0.0007 | −0.0065 | 0.1111 | −2 |
| ALC (Subject 2) | −0.001 | 0.0087 | −0.0202 | −0.0637 | 2.2 |
| PLC (Subject 2) | 0.0047 | −0.0368 | 0.0826 | 0.1034 | −1.99 |
| ALC (Subject 3) | 0.0005 | −0.0046 | 0.0155 | −0.0715 | 2.09 |
| PLC (Subject 3) | 0.0037 | −0.0307 | 0.0715 | 0.096 | −2.1 |

ALC: anterior lens curvature; PLC: posterior lens curvature.

**Table 2.** Characteristic parameters of personalized lens finite element model.

| Parameter | Subject 1 | Subject 2 | Subject 3 |
|---|---|---|---|
| Ciliary body radius, $R_{cb}$ [mm] | 6.465 | 6.492 | 6.51 |
| Lens radius, $R_L$ | 4.32 | 4.3 | 4.28 |
| Nucleus thickness, $t$ [mm] | 2.97 | 2.6 | 2.69 |
| Capsule Young's modulus [N/mm²] | 1.3 | 1.2 | 1.15 |
| Nuclear Young's modulus [N/mm²] | 0.00055 | 0.00055 | 0.00055 |
| Cortex Young's modulus [N/mm²] | 0.00342 | 0.00342 | 0.00342 |
| Anterior zonule stiffness [N/mm] | 0.066 | 0.066 | 0.066 |
| Central zonule stiffness [N/mm] | 0.011 | 0.011 | 0.011 |
| Posterior zonule stiffness [N/mm] | 0.033 | 0.033 | 0.033 |
| Capsule Poisson's ratio | 0.47 | 0.47 | 0.47 |
| Nuclear and cortex ratio | 0.49 | 0.49 | 0.49 |

## 3. Results

As mentioned above, all parameters required for the personalized lens finite element model were obtained by only one SS-OCT scan under the maximum accommodation state. At this time, the tensile distance of the model (*Dc*) is 0. The image processing method introduced in Section 2.3 was used to extract the curvature radius of the anterior and posterior surfaces of the lens, the lens thickness, and the lens nucleus. Figure 4 shows the mesh deformation diagram and displacement program of the lens at different tensile distances. As the suspensory ligament pulls on the lens by the ciliary body, the lens becomes thinner and the surface profile becomes more gradual. Figure 4a–c are the mesh deformation diagrams when Dc is equal to 0, 0.2, and 0.36, respectively. Figure 4d–f is the displacement nephograms under the corresponding tensile distances. When the tensile distance Dc is equal to 0.36, the lens can be considered to be in the no-accommodation state.

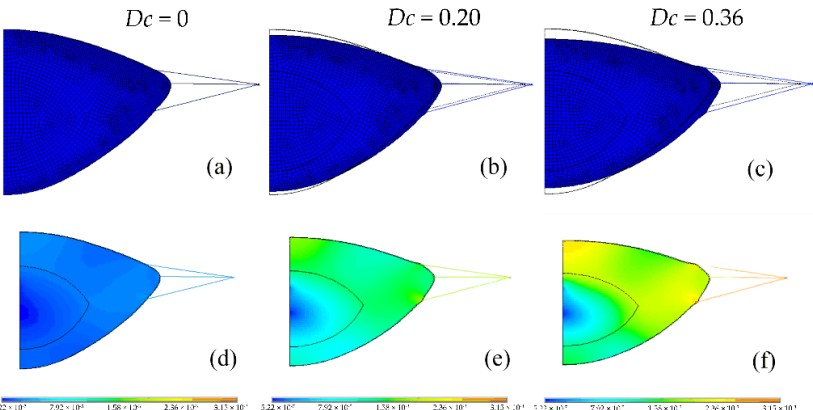

**Figure 4.** (**a–c**) Mesh deformation diagram of lens finite element model under different tensile distances; (**d–f**) The corresponding displacement nephogram.

With each pulling of the ciliary body, the five-order polynomial based on the least squares method was repeatedly used to describe the contour of the lens model and calculate the model parameters at this moment. Figure 5 shows the variation of the finite element model parameters of three subjects with tensile distance.

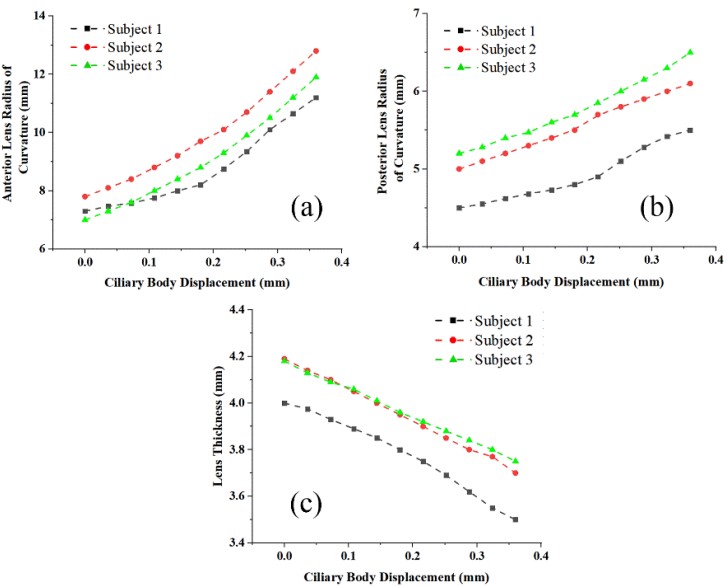

**Figure 5.** Variations of geometric parameters of lens finite element model with ciliary body pulling in 3 subjects: (**a**) curvature radius of anterior surface; (**b**) curvature radius of posterior surface; (**c**) lens thickness.

Figure 5a,b show the changes in the center curvature radius of anterior and posterior lens surfaces with tensile distance, respectively. Figure 5c shows the change of lens thickness with ciliary body pulling. It also shows the parameters of three subjects separately, which more intuitively showed the individual differences between the subjects. It can be seen from the figure that the posterior surface curvature and lens thickness of subject 1 are significantly different from the other two subjects. After parameter analysis of three subjects, it can be noted that the average curvature radius of the anterior and posterior surfaces varies by 4.6 mm and 1.13 mm in the pulling range of 0~0.36 mm in the ciliary body and the average lens thickness change was 0.473 mm. According to the existing statistical results of big data, the healthy subjects lens posterior surface curvature range is between 4–7 mm, and the lens

thickness range is between 3–5 mm. Therefore, the result is within the normal range.

In particular, because there are great individual differences in the eye parameters of each person, the general model is not applicable to everyone, even if the subjects are healthy people. This further indicates the importance of personalized human lens model.

As the tensile distance of the ciliary body increases, the lens model was transformed from the maximum-accommodation state to the no-accommodation state, and its refractive power decreases significantly, as shown in Figure 6a. To show the variation of refractive power with the tensile distance more clearly, we introduce the concept of lens accommodative response. The lens accommodative response is defined as "the refractive power of the lens after the accommodation stimulus" minus the "the refraction power of the lens at no-accommodation state". Figure 6b shows the change of lens accommodative response with tensile distance. The total accommodative response of the finite element model is about 6.95 D.

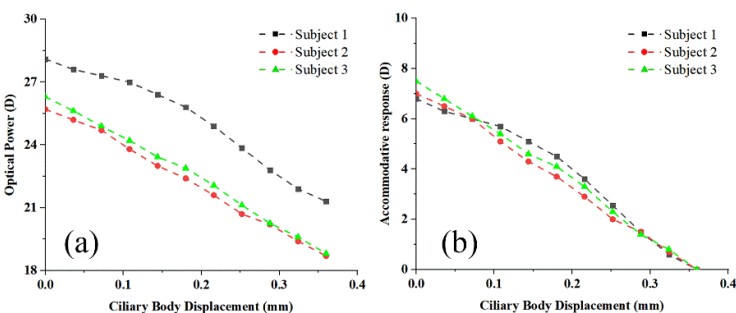

**Figure 6.** Variations of refractive power (**a**) and accommodative response (**b**) with ciliary body pulling in 3 subjects.

In the above, we established the finite element model using the lens parameters at the maximum-accommodation state, and simulate the changes in shape, refractive power, and the accommodative response of the lens caused by accommodation. Next, we used SS-OCT to collect the shape, refraction power, and accommodative response changes of the real human lens under different accommodation stimulus. Figure 7a–i shows the anterior segment images of three subjects under the accommodation stimulus of 0 D, 3 D, and 9 D, respectively. Figure 7j,k show comparisons in different accommodation states of a subject. The left side of the two sub-images is both 0 D stimulus, and the right side is 3 D and 9 D stimulus, respectively. It was found that the anterior and posterior surfaces of the lens became steeper and the lens thickness also increased with the enhancement of the accommodation stimulus.

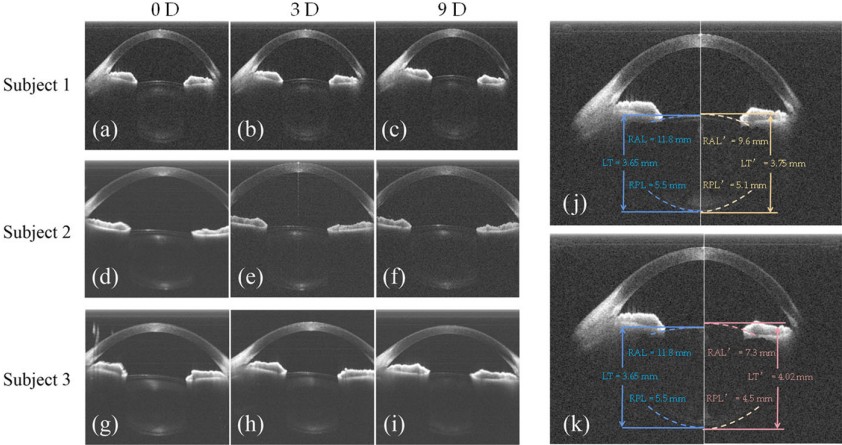

**Figure 7.** Comparison of anterior segment images obtained by SS-OCT. (**a–i**) the anterior segment images of three subjects under the accommodation stimulus of 0 D, 3 D, and 9 D, respectively. (**j**)

and (**k**) the comparisons in different accommodation states of a subject. The left side of the two sub-images is both 0 D stimulus, and the right side is 3 D and 9 D stimulus, respectively.

Figure 8a,b show the curvature radius of the anterior and posterior surfaces of the real lenses in three subjects as a function of the accommodation stimulus. The mean curvature radius of the anterior surface was 11.8 ± 0.5 mm and that of the posterior surface was 5.8 ± 0.3 mm at 0 D stimulus. The average changes in the curvature radius of the anterior and posterior surfaces were 0.495 mm/D and 0.11 mm/D, respectively. However, there was no significant change when the accommodation stimulus was above 7 D, which may be due to the maximum accommodative response limit by the subjects. Figure 8c shows the change in lens thickness of the three subjects. The average lens thickness change was 41.5 μm/D.

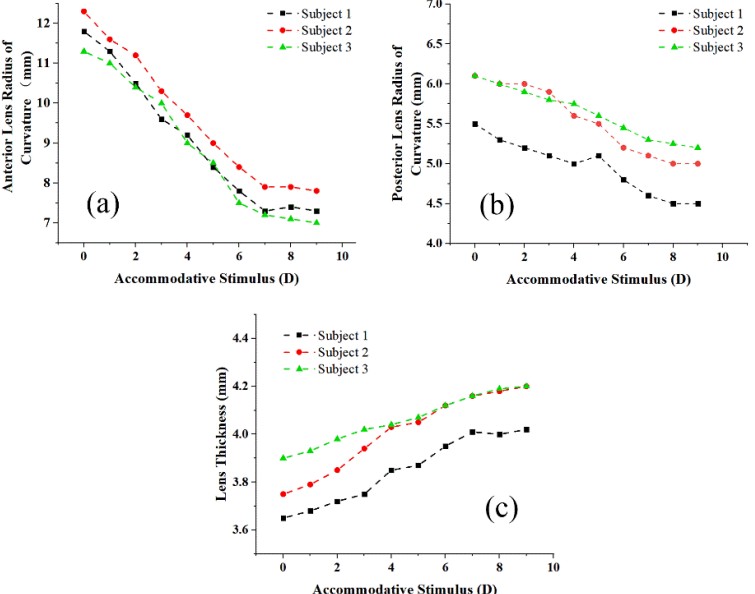

**Figure 8.** The change of lens geometric parameters during accommodation in 3 subjects: (**a**) anterior lens curvature radius; (**b**) posterior lens curvature radius; (**c**) lens thickness.

Figure 9a shows the refractive power of the lens as a function of the accommodative stimulus. The overall refractive power of the lens ranges from 19.5 D to 28.1 D. The average variation of lens refractive power was 0.74 D/D. This is equivalent to 0.74 D response /D accommodating demand. The total change from no-accommodation state to maximum-accommodation state was about 6.7 D. Similarly, when the accommodative stimulus reached 7 D, the changes in the refractive power of the lens ceased to be significant.

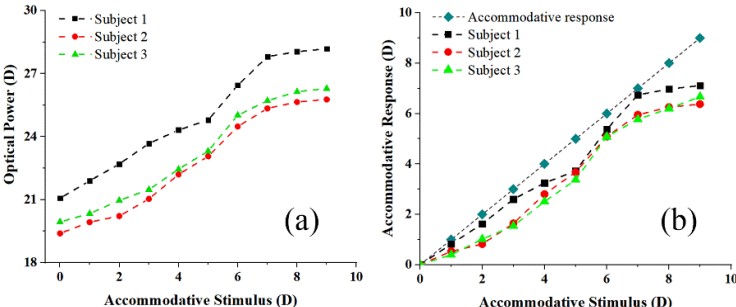

**Figure 9.** Variations of refractive power (**a**) and accommodative response (**b**) with accommodation stimulus in 3 subjects.

In order to verify the accommodation accuracy of the personalized lens finite element model more directly, the accommodation process of the finite element model and real lens were drawn in the same picture for comparison. The accommodation process data of the finite element model were reversed to make it change from the no-accommodation state to the maximum-accommodation state. This matched the real lens accommodation process. The abscissa of the finite element model accommodation process was changed from "0~0.36" to "0.36~0" and placed above the curve. The abscissa of the real lens accommodation process remains unchanged (0~9 D) and was placed below the curve, as shown in Figure 10.

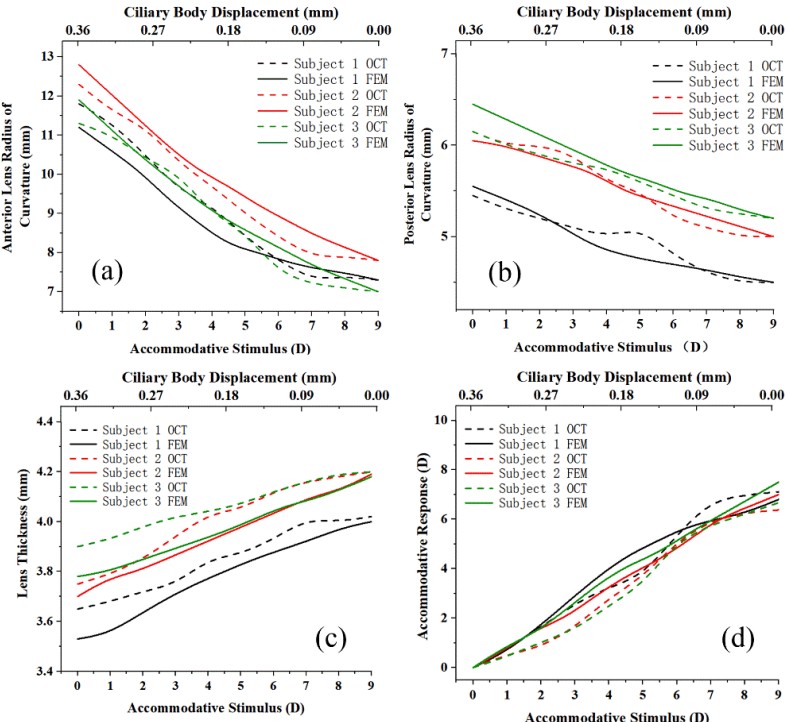

**Figure 10.** The verification of accommodation accuracy of personalized lens finite element model. Comparison of four parameters between the finite element model and the real lens in the process from the no-accommodation state to the maximum-accommodation state. (**a**–**d**) are curvature radius of the anterior and posterior surfaces, thickness, and the accommodative response of the lens, respectively. FEM: finite element model. OCT: Optical Coherence Tomography.

As can be seen from Figure 10, the accommodation processes of the finite element model are highly consistent with that of the real lens. This proves that the personalized lens finite element model proposed in this paper can accurately simulate the real human eye lens accommodation process. Table 3 shows the average difference and the difference rate (average difference/mean of parameters) between the two. The values outside the brackets represent the average difference, and the values inside the brackets represent the difference rate.

**Table 3.** Average difference and difference rate between the accommodation process of finite element model and real lens (average difference/mean of parameters).

| Subject | ALC | PLC | TH | RP |
|---|---|---|---|---|
| Subject 1 | 0.38 (4.5%) | 0.10 (2.0%) | 0.067 (1.8%) | 0.45 (1.8%) |
| Subject 2 | 0.31 (3.3%) | 0.07 (1.3%) | 0.058 (1.5%) | 0.36 (1.6%) |
| Subject 3 | 0.25 (2.8%) | 0.12 (2.2%) | 0.09 (2.3%) | 0.59 (2.5%) |
| Average | 0.313 (3.53%) | 0.096 (1.83%) | 0.071 (1.86%) | 0.466 (1.96%) |

The values outside the brackets represent the average difference, and the values inside the brackets represent the difference rate. ALC: anterior lens curvature; PLC: posterior lens curvature; TH: lens thickness; RP: refractive.

As can be seen from Table 3, the difference rate of all parameters is less than 5%. The difference in anterior surface curvature of the lens was the largest, with a maximum difference rate of 4.5% and an average of 3.53%. The difference rate of the posterior surface curvature, lens thickness, and lens refraction power were no more than 2.5%. Their maximum difference rates were 2.2%, 2.3%, and 2.5%, with the mean values of 1.83%, 1.86, and 1.96%, respectively.

## 4. Discussion

Accurate reproduction of the lens accommodation process of a specific individual has a guiding role in exploring the physiological and pathological mechanism of presbyopia and cataract-related diseases, searching for effective treatment methods, optimizing the mechanical design of the iOL and developing the iOL with regulatory function. The lens model established by Burd et al. has been widely adopted by scholars because it can describe most of the lens behaviors in the accommodation process [15,24,62]. Later, Hooman et al. improved Burd's model [63]. The accuracy of every model is dependent on the correctness of the data used to generate it, according to the knowledge gained from earlier finite element modeling. The precision of the geometric parameters and the material properties necessary for the lens model are key factors in the creation of a customized lens model, the accuracy of which is largely dependent on these factors. The SS-OCT-based personalized finite element model proposed in this paper was realized by obtaining accurate lens geometric parameters through SS-OCT, adjusting and optimizing the Burd classic 29-year-old model. The specific content includes: 1. In order to improve the accuracy of the model, a quadrilateral eight-node element was adopted instead of the original triangular element; 2. Data density was increased to realistically reproduce the lens accommodation process; 3. To boycott the contact body slide or separation, the adhesion between the layers was increased. The three face-to-face nodal contact units were arranged at the boundary of contact between the nucleus, cortex, and capsule.

During the real lens accommodation process, the change in the lens parameter based on SS-OCT is slightly smaller than those obtained by Dubbelman et al. (0.61 mm/D and 0.13 mm/D) [58]. This is because we applied a higher accommodation stimulus (0 D to 9 D). However, when the accommodative stimulus reached 7 D, the changes in the refractive power of the lens ceased to be significant. The changes in lens thickness obtained by SS-OCT were similar to the results measured by Pablo et al. (40 μm/D) [64] but less than the results measured by Doyle et al. (57 μm/D) [65]. The lens refractive power and accommodative response changes obtained in this paper have a similar trend to the 0.69 D/D, as described by Eduardo et al. [66], which is lower than the 0.81 D/D obtained by simulating the 3D surface profile of the lens by Pablo et al. [64]. This may be due to individual differences.

Although the accommodation process of the finite element model is highly consistent with that of the real lens, there are still some numerical differences. This may be due to the atypical accommodative responses of the accommodation stimulus, or to the fact that ciliary body stretching does not correspond linearly to the accommodation stimulus of the real human eyes. In addition, the division of the lens nucleus is also a factor affecting the accuracy of the finite element model. Fisher [67] and Wilde [68] et al. found that the elastic modulus of the young lens nucleus was larger than that of the lens cortex, resulting in a higher deformation of the lens cortex with stretching than that of the nucleus. This suggests that small nuclear divisions may lead to greater deformation in cortical regions than in reality. The development of lens finite element modeling and human eye imaging technology is promising, with regard to its ability to solve the above problems.

## 5. Conclusions

SS-OCT can provide complete geometric parameters for finite element lens modeling. The personalized lens finite element model almost accurately reproduced the accommodation process of the real lens. We provide a new method for the study of the human eye accommodation mechanisms. This work will promote the development of iOL design.

**Author Contributions:** Conceptualization, Z.M. and J.L. (Jian Liu); methodology, G.L.; software, A.L. and J.L. (Jingmin Luan); validation, Y.Z. and K.Z.; formal analysis, S.Y. and Z.L.; investigation, H.L.; resources, Y.L. and S.W.; data curation, Y.Y. and Y.W.; writing—original draft preparation, A.L.; writing—review and editing, G.L.; visualization, J.L. (Jian Liu); supervision, J.L. (Jian Liu); project administration, Z.M.; funding acquisition, Z.M. All authors have read and agreed to the published version of the manuscript.

**Funding:** This research was funded in part by National Natural Science Foundation of China, grant number 61771119, 61901100 and 62075037, Hebei Provincial Natural Science Foundation of China, grant number F2019501132, E2020501029, and F2020501040.

**Institutional Review Board Statement:** The research followed the tenets of the Declaration of Helsinki and was approved by the Ethics committee of Northeastern University (No. Neu-ec2020a009s).

**Informed Consent Statement:** Informed consent was obtained from all subjects involved in the study.

**Data Availability Statement:** Data underlying the results presented in this paper are not publicly available at this time but may be obtained from the authors upon reasonable request.

**Conflicts of Interest:** The authors declare no conflict of interest.

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
