# Peer review of "Establishment of Personalized Finite Element Model of Crystalline Lens Based on Sweep-Source Optical Coherence Tomography"

_photonics, doi:10.3390/photonics9110803_

Round 1

Reviewer 1 Report

General

The study approaches a FEM of the crystalline lens using SS-OCT. The study is rigorous, the paper is sound, while OCT is a hot topic. The paper can therefore be considered for publication in Photonics, with some improvements, as pointed out bellow.

Specific

1) The English has some issues, with edits here and there, for example “slightly small than” should be “slightly smaller than” (Line 361), etc. Please recheck the entire manuscript in this respect.

2) Please provide for readers the lateral resolution of the utilized SS OCT system, as well as other characteristic, such as penetration depth.

3) Please provide more initial refs on OCT when introducing the technique, including:

the first study introducing OCT

Huang, D.; Swanson, E.A.; Lin, C.P.; Schuman, J.S.; Stinson, W.G.; Chang, W.; Hee, M.R.; Flotte, T.; Gregory, K.; Puliafito, C.A.; et al. Optical coherence tomography. Science 1991, 254, 1178–1181.

a ref on advantages and characteristics of SS-OCT,

Drexler, W.; Liu, M.; Kumar, A.; Kamali, T.; Unterhuber, A.; Leitgeb, R.A. Optical coherence tomography today: speed, contrast, and multimodality. J Biomed Opt 2014, 19, 071412.

as well as on the maximum resolution of OCT

Cogliati, A., Canavesi, C., Hayes, A., Tankam, P., Duma, V.-F., Santhanam, A., Thompson, K. P., and Rolland, J. P., MEMS-based handheld scanning probe with pre-shaped input signals for distortion-free images in Gabor-Domain Optical Coherence Microscopy, Opt. Express 24(12), 13365-13374 (2016).

Also, please justify for the readers the choice of the OCT imaging technique for this research.

4) Please correct figures to make sure that inscriptions have the same size. Also, several Figs., such as 5 to 10 are not fine, regarding format and legends. Figure 7 j and k is totally not readable. Please adjust.

5) The number of the Ethical Approval must be provided.

6) Please point out the reason for choosing three subjects for the study. Is this group enough? What about statistics? This is the major issue of the study.

7) Regarding the previous point, what about all the figs with the three graphs – each for one of the subjects? A more detailed discussion on differences between them should be provided, especially as all three are selected as healthy, etc.

8) Please remove self-assessments for this paper, such as “This work is of great significance” in Line 384. Please check the entire text for such expressions and correct.

Reviewer 2 Report

The manuscript submitted by Guangheng Liu et al. is concerning the new approach to develop the  finite element model for the crystalline lens based on the OCT measurement. Even if the use of a finite element model is not a novel approach, as these techniques have been used for several years to model the optical surfaces of the human eye, the use of SS-OCT data in these models is an interesting approach in line with popular recent trends focused on personalised medicine. The issues addressed in the manuscript overlap with the scope of the Photonics journal. The language used in the manuscript should be understandable to a wide range of readers.  However, some corrections and clarifications must be made before publication.

The selection of literature does not raise any major objections to me, although the authors should also refer to items in the literature directly related to the modelling of the optical surfaces of the eye using the finite element method. Important references are missing from the current version.

The logical structure and organization of the manuscript do not raise any major concerns for me.  However, as far as I remember, in this journal it is obligatory to have a section referring to the conclusion, which is not present in the current version of the article. Please correct this.

The results were carried out on only three patients, which is not a very large number. Do the authors believe that the results obtained are representative?

Did the authors measure the lens of only one eye or both eyes when examining patients with OCT?

The manuscript does not say how many times OCT measurements were repeated in the three patients for each accommodative stimulation. Was it a single measurement or multiple measurements. This information should appear in the manuscript, as the results from a single measurement definitely cannot be representative.

Lines 163-162: There is no description of the OCT data processing algorithms used. Provide detailed and relevant information.

Can the authors provide more data on the patients studied, as the results obtained show that in the case of patient 1, his results presented in Fig.5b,c/Fig.6/Fig.8  differed from the results obtained for the other two patients. What was the reason for this?

The manuscript does not indicate whether the OCT system used was a commercial system or whether it was constructed in the group carrying out this research. If it was a commercial system, the authors should provide the model name, manufacturer, and basic technical data as required by the journal.

The manuscript contains typos and editing errors, e.g.  see line 114: “modle”? , missing spaces, missing colons (see, e.g., lines 353-359), etc. I recommend carefully reading the manuscript and correcting these errors.

All abbreviations appearing in the text should be explained.

Round 2

Reviewer 1 Report

The paper has been substantially revised, according to all comments and suggestions. In the opinion of this reviewer, the manuscript can be accepted in the present form for publication in Photonics.

Reviewer 2 Report

I would like to thank the authors for their replies and for the corrections made. In my opinion, the current version of the manuscript is suitable for publication.